# A framework to identify opportunities to address socioscientific issues in the elementary school curricula: A case study from England, Italy, and Portugal

Patrícia Pessoa[1]*, Joelyn de Lima[2], Valentina Piacentini[3], Giulia Realdon[4], Alex Jeffries[5], Lino Ometto[6], J. Bernardino Lopes[7], Dana L. Zeidler[8], Maria João Fonseca[9], Bruno Sousa[10], Alexandre Pinto[11], Xana Sá-Pinto[12]

**1** Research Centre in the Didactics and Technology in the Education of Trainers (CIDTFF.UA), Department of Education and Psychology, University of Aveiro, Aveiro, Portugal; and University of Trás-os-Montes e Alto Douro (UTAD), Vila Real, Portugal, **2** W.K. Kellogg Biological Station, Michigan State University, E Gull Lake Dr, Hickory Corners, Michigan, United States of America and The Swiss Federal Institute of Technology (EPFL), Lausanne, Switzerland, **3** Research Centre in the Didactics and Technology in the Education of Trainers (CIDTFF.UA), Department of Education and Psychology, University of Aveiro, Aveiro, Portugal; and "Via Merope" school cluster, Rome, Italy, **4** UNICAMearth Group, Geology Section, University of Camerino, Italy, **5** Milner Centre for Evolution, Department of Life Sciences, University of Bath, Claverton Down, United Kingdom, **6** Department of Biology and Biotechnology, University of Pavia, Pavia, Italy, **7** Research Centre in the Didactics and Technology in the Education of Trainers (CIDTFF.UA), Department of Education and Psychology, University of Aveiro, Aveiro, Portugal; and University of Trás-os-Montes e Alto Douro (UTAD), Vila Real, Portugal, **8** Department of Teaching and Learning, College of Education, East Fowler Avenue, University of South Florida, Tampa, Florida, United States of America, **9** Natural History and Science Museum of the University of Porto (MHNC-UP), Praça Gomes Teixeira, Porto, Portugal, **10** Alpoente - Albufeira poente school cluster, Rua das Escolas, Apartado , Albufeira, Portugal, **11** Centre for Research and Innovation in Education - inED, ESE, Polytechnic of Porto, Porto, Portugal, **12** Research Centre in the Didactics and Technology in the Education of Trainers (CIDTFF.UA), Department of Education and Psychology, University of Aveiro, Aveiro, Portugal

* afppatricia@gmail.com

## Abstract

Scientific literacy is crucial to address important and complex societal problems, both current and future. Teaching using a socioscientific issues (SSI) approach is a potential strategy to develop students' scientific literacy, although teachers have reported concerns about its implementation, such as the inability to add additional distinct requirements to already demanding curricula. To facilitate this task, we describe the development of a valid and reliable instrument for curricula analysis, called "Framework for Identifying Opportunities to implement an SSI approach in science school curricula" (FIOSSI), and use it to identify opportunities to implement the SSI approach in the elementary school science curriculum of three European countries (England, Italy and Portugal). The framework categorizes SSI opportunities into three areas: 1) awareness of the issue; 2) socioscientific reasoning; and 3) socioscientific identity. Our analyses of the three countries' elementary curricula reveal that the current versions have significant opportunities to explore awareness of SSI (especially relating to environmental and human health issues), promote the development of socioscientific reasoning, and foster socioscientific identity. FIOSSI can be a useful tool for education research and practice, and our results can help inform future research and guide educational policies.

**Data availability statement:** The learning goals considered to be aligned with any of the categories of FIOSSI, together with the final consensus coding can be found in the dataset available at https://doi.org/10.48527/NN0MJC

**Funding:** P.P., A.P. and X.S.P. are funded by Portuguese national funds through FCT – Fundação para a Ciência e a Tecnologia, I.P. (https://www.fct.pt/), within the scope of the PhD grant 2020.05634.BD (https://doi.org/10.54499/2020.05634.BD), the project UIDP/05198/2020 (Center for Research and Innovation in Education, inED, https://doi.org/10.54499/UIDP/05198/2020), and the Scientific Employment Stimulus - Individual Call – 2022.06070.CEECIND/CP1720/CT0041 (https://doi.org/10.54499/2022.06070.CEECIND/CP1720/CT0041), respectively. This work is also financially supported by Portuguese national funds through FCT – Fundação para a Ciência e a Tecnologia, I.P. (https://www.fct.pt/), under the CIDTFF projects UIDB/00194/2020 (https://doi.org/10.54499/UIDB/00194/2020) and UIDP/00194/2020 (https://doi.org/10.54499/UIDP/00194/2020). The funders had no role in study design, data collection and analysis, decision to publish, or preparation of the manuscript.

**Competing interests:** The authors have declared that no competing interests exist.

## Introduction

The world is facing an increasing number of pressing challenges that require a science-based response. To address the various social, economic and environmental challenges affecting the world, the United Nations [1] established 17 Sustainable Development Goals (SDGs) to achieve the 2030 Agenda for Sustainable Development. The 2023 mid-point report rang alarm bells as it indicated that progress on most of the goals was moderately to severely off track, and a significant number had regressed below the 2015 benchmarks [2]. A scientifically literate populace plays a pivotal role in promoting awareness, understanding and action related to the SDGs. Therefore, it is imperative that we ensure that our students are scientifically literate [3]. Our young students should be afforded opportunities to develop competencies that allow for the making of informed decisions, develop innovative and effective solutions, and actively engage with debates related to local and global problems. Educators have a prima facie responsibility for providing quality education (SDG4) that is fundamental to develop students' key competencies in sustainability, and empowering them with knowledge, skills and attitudes needed to successfully address complex issues and ultimately achieve all other SDGs [4,5].

Most of today's sustainability challenges entail complex, ill-structured, and controversial societal issues, linked to scientific ideas and principles that lack simple solutions, and are potentially informed by concepts from multiple fields that have economic, political, and ethical implications [6,7]. In other words, most of the sustainability challenges are Socioscientific Issues (SSI) [8]. Thus, responsible sustainable development requires science education to develop public scientific literacy, allowing people to engage, as reflective citizens, in science-related issues [9,10].

Scientific literacy is described by Roberts [11] through two visions: while Vision I is limited to science content knowledge, Vision II broadens the focus and proposes situating science in everyday sociocultural contexts. Building from Robert's Vision II, some authors have proposed Vision III, which further extends the concept and aims to promote competencies that allow students to reflect and discuss science within the scope of socio-political problems, addressing social justice and ethical considerations through social activism and awareness, and facilitating moral reasoning [12–15].

The Socioscientific Issues (SSI) educational framework is directly aligned with Vision II and Vision III [8]. This approach is influenced by the Science, Technology and Society (STS) tradition and is further informed by scholarship from philosophical, developmental, and sociological traditions [7]. As a science education approach, SSI has four key elements [8]: 1) use personally relevant, controversial, and ill-structured problems that require scientific, evidence-based reasoning to inform decisions; 2) employ the use of scientific topics with social implications that require students to engage in dialogue, discussion, debate, and argumentation; 3) integrate implicit and/or explicit ethical components that require moral reasoning; and, 4) emphasize the development of virtue and character as long-range pedagogical goals.

To support the design and implementation of the SSI approach, Sadler et al. [16] proposed a model for SSI teaching and learning outlining key learning objectives such as awareness of the issue, socioscientific reasoning, and identity development (among others, such as scientific practices and the nature of science, which are not exclusive to this approach, as well as crosscutting concepts and core ideas, which are specific to the context of the USA). Awareness of the issue involves understanding how scientific concepts relate to the issue and recognizing associated social problems [reviewed in [16]. Socioscientific reasoning includes understanding the complexity of SSI, analyzing issues from multiple perspectives, identifying aspects subject to ongoing inquiry, employing skepticism toward potentially biased information, and exploring science's contributions and limitations regarding the issue [16]. Multiple studies [17,18] highlight the complexity of SSR and proposed the need for sustained efforts, as opposed to short-term interventions, to effectively influence students' SSR. Identity development

encourages students to see themselves as capable contributors to SSI discussions, engage with these issues both in and out of school, and develop new competencies and interests [reviewed in [16,19,20]. However, identity develops over time [21] and can evolve through extended engagement with meaningful issues [6]. Therefore, it is important to consider how students' identities change through multiple SSI learning experiences over the years [16].

The SSI approach has indeed been shown to promote the development of students' understanding of the nature of science [22,23]; students' reasoning [24,25] and argumentation skills [26,27]; functional scientific literacy and character [28,29]; and moral reasoning and sensitivity [30–34]. SSI has also been used to foster systems thinking, anticipatory, normative, strategic, and interpersonal competencies in students [35]. This highlights the potential of this approach to foster the key competencies in sustainability, needed to empower people with knowledge, skills and attitudes essential to successfully address complex issues, and to ultimately achieve all SDGs [4,5].

While there are some promising studies that have shown SSI-based education to be particularly relevant for elementary school students connecting science knowledge with social issues, thereby providing a foundation for more complex reasoning and understanding of scientific concepts [36–39], most of the research and resources in the extant literature tend to be focused on higher education, not on elementary school levels [36–39]. Additionally, despite evidence supporting the educational potential of exploring SSI in the classroom, teachers report lack of confidence [37,40] and several concerns related to using this educational approach, including: *i)* teaching SSI often demands the exploration of unfamiliar topics and knowledge outside their immediate content domain (e.g., ethical/moral, political, economic, sociocultural issues) [36,37], *ii)* lack of suitable teaching materials [41,42], *iii)* concerns about students' knowledge and understanding ability [37], *iv)* time constraints in designing and implementing SSI-based lessons [37], and *v)* intolerance of uncertainty [40]. Despite the model proposed by Sadler et al. [16] for SSI teaching and learning, which aimed to assist teachers in designing and implementing this approach, allocating sufficient time for in-depth exploration of SSI remains a challenge for teachers due to the current demands of school curricula [37,41]. School curricula are a set of official policy documents, created by the relevant ministries of education and/or other state or local authorities [43], and represent "the expression of educational ideas in practice" (44, p. 326). For the students of a particular country/state/locality, school curricula define standards regarding the achievement of learning goals and the development of skills and competencies. Curricula may vary extensively in both type and structure [44]. While social mores and changes influence curricula content and priorities, curricula in turn, can affect social change [45] as they comprise standards that students are compelled to meet. As teachers are faced with having to cover a wide range of topics and address a wide range of competencies, they may face pedagogical pressures [46], making it challenging for teachers to allocate sufficient time for concepts or approaches not explicitly defined in the curriculum [37,40,41]. In contrast, some authors have argued that the SSI framework can serve as a value-added approach to not only engage students in the activity of science but can also serve as an anchoring point to subsume and scaffold other subject matter from multiple disciplines [47–49].

To the best of the authors' knowledge, to date, no study has investigated opportunities to implement SSI at the elementary school level in European science school curricula. To overcome this lack of knowledge, in this study we aimed to:

1) develop a framework to conduct curriculum content analysis related to SSI; and,

2) apply it to characterize the elementary school science curricula of three European countries (England, Italy and Portugal) regarding the opportunities to address different dimensions of the SSI approach.

Based on our findings, we examine the contributions to educational research and practice, discuss their relevance to educational policy and curriculum development and their implications for elementary teacher training and further research.

We argue that the identification of opportunities to implement SSI at the elementary school level may help reduce the constraints felt by teachers. By clarifying and highlighting existing opportunities, we aim to promote the integration of the SSI approach into the current curriculum rather than perceiving it as an additional task. Additionally, the identification of these opportunities may enhance the future development of educational activities and materials and inform practitioners about the degree of alignment of European science curricula with Visions II and III of scientific literacy.

## Methods

### Sample

Diverse curriculum designs and traditions exist across Europe, varying from flexible structures in some countries to quite extensive in others, with very detailed descriptions of the concepts that teachers should focus on, and of the learning goals to be achieved by students, and in some cases, even the educational methodologies to achieve them [45]. To develop the analysis framework, we examined the official curriculum of England [50], Italy [51] and Portugal [52] for the 2021-2022 academic year. These curricula constitute a convenience sample [53]: the researchers on the team were from these three countries and therefore were familiar with the respective curricular documents and were interested in their analysis [53]. We focused on the analysis of elementary school science curricula, which share common features, such as the inclusion of learning objectives and suggestions for teaching strategies, yet show wide variation in terms of organization (see Table 1) and flexibility provided to teachers and disciplines. Besides the learning goals that the students are expected to achieve, these documents also provide guidelines for educational approaches to be implemented by the teachers. According to the national regulations in Italy and Portugal, textbooks need to comply with the objectives, contents, and guidelines in the official curriculum [Italy: [54]; Portugal: [55,56]. In England, there is no specific legislation requiring textbooks to meet curriculum standards; however, publishers typically ensure alignment to support schools in meeting these requirements. The sections of the curriculum for each country used to develop the analysis were as follows:

1) English school curriculum (EN): This curriculum is organized in key stages and further divided by school grades, including statutory requirements and optional suggestions [57]. The English elementary school system encompasses grades 1 to 6, and compulsory education begins at the age of 5, spanning a total of 13 years. We analyzed the science subject of the elementary school curriculum from grades 1 to 6 (from 5 to 10 years old). The document that specifies learning goals up to grade 6 has 37 pages.

2) Italian school curriculum (IT): This curriculum has a two-stage (age span) basis. The Italian elementary school system covers grades 1 to 5 (from 6 to 10 years old), the learning objectives being presented as: objectives that students should acquire by the end of grade 3 (first stage) and by the end of grade 5 (second stage) [51]. This two-stage curriculum structure allows schools and teachers to organize the teaching of the prescribed topics across the various grades included within each stage. Compulsory education starts at the age of 6, extending for a total of 10 years. We analyzed the science subject of the elementary school curriculum up to grade 5. Grade 6 was not included in this analysis because it is part of the next education stage, which is treated as a single block in the curriculum documents (lower secondary school, grades 6 to 8). The document that guides the learning goals up to grade 5 has 4 pages.

**Table 1. Overview of the structure of each curriculum.**

| England [50] | Italy [51] | Portugal [52] |
|---|---|---|
| Overall curriculum | Introduction (general aims) | General aims for grades 1 to 4 (6 to 9 years old) |
| General aims | Targets for the development of competencies at the end of elementary school | |
| Key stage 1 (learning goals for grades 1 and 2 - 5 to 6 years old) | Learning goals for grades 1 to 3 (6 to 8 years old) | Grade 1 (6 years old) |
| Grade 1 (5 years old) | | |
| Grade 2 (6 years old) | | Grade 2 (7 years old) |
| Lower key stage 2 (learning goals for grades 3 and 4 - 7 to 8 years old) | | Grade 3 (8 years old) |
| | Grade 3 (7 years old) | |
| Grade 4 (8 years old) | Learning goals for grades 4 to 5 (9 to 10 years old) | Grade 4 (9 years old) |
| Upper key stage 2 (learning goals for grades 5 and 6 - 9 to 10 years old) | | General aims for grades 5 to 6 (10 to 11 years old) |
| Grade 5 (9 years old) | | Grade 5 (10 years old) |
| Grade 6 (10 years old) | | Grade 6 (11 years old) |

3) Portuguese school curriculum (PT): This curriculum prescribes essential learning goals on a per-grade basis at the national level, while providing curricular autonomy to teachers and school clusters, to explore interdisciplinary articulations, and local components of the curriculum [58]. Basic education is divided into three cycles: the first cycle encompasses 4 years (grades 1-4), the second cycle covers 2 years (grades 5-6), and the third cycle spans 3 years (grades 7-9). Compulsory education in Portugal starts at the age of 6 and lasts for 12 years. We analyzed the curriculum up to grade 6 (11 years old). From grade 1 to 4 (from 6 to 9 years old), we analyzed the curriculum for the subject "Study of the Environment" that encompasses Biology, Physics, Geography, Geology, History, Chemistry, and Technology. For grades 5 to 6 we analyzed the curriculum of the subject "Natural Sciences". The documents that guide the learning goals up to grade 6 sum 64 pages.

While there may be relevant SSI-related learning goals in subjects like History or Technology, our analysis focused solely on goals taught within subjects in the field of natural sciences. Although it is not the aim of this study to compare the curricula of the three countries, but rather to highlight the opportunities for addressing socioscientific issues within each curriculum individually, we focused both on comparable grade ranges, in terms of years of schooling and students' age, and teacher training. In England, Italy and Portugal, the training requirements for elementary school teachers differ from those for teachers at higher educational levels. In England, a specific degree in elementary education, such as a Bachelor of Education (BEd) or a Postgraduate Certificate in Education (PGCE) focused on elementary education, is required to teach all subjects in elementary school (grades 1-6) [59]. In Italy, a specific master's degree in elementary education (Sciences for teaching in elementary education) is required to teach all subjects in elementary school (grades 1-5) [60,61]. In Portugal, a specific master's degree in Basic Education is required to teach all subjects in grades 1-4 and to teach subjects such as Science and Mathematics or History and Portuguese in grades 5-6 [62].

## Content analysis

**Units of analysis.** We applied a content analysis [63] to the three school curricula. The "meaning unit" as the unit of analysis, refers to a collection of words or statements that pertain to a common thread or central meaning [64]. In this study, these meaning units consisted

of curriculum learning goals or their relevant segments that aligned with a specific category, subcategory or sub-subcategory of the analysis framework.

**Coders' profiles and expertise.** The coders of this research are the authors of this paper and possess diverse professional and educational backgrounds (junior and senior researchers in science education, as well as science teachers), as needed to ensure the reliability of the analysis [63]. To enhance reliability of data analysis, each curriculum was analyzed by a minimum of two coders who were fluent in the appropriate language, independently examined the documents and identified and coded any SSI-related goals mentioned in those documents [63]. The coders were involved in the development of the framework of analysis and afterwards in the final step of identification and coding of the units of analysis.

## Framework development

To develop FIOSSI, we identified learning objectives that can be explored under the SSI framework, and competencies that can be fostered under this approach. From the learning objectives referred by Sadler et al. [16], we decided to further include in our instrument of analysis the learning objectives *awareness of the issue*, *socioscientific reasoning* and *socioscientific identity*, and to convert them into categories of analysis. The remaining objectives were not included in the present analysis as these were not specific features of the SSI approach (as it happens with *scientific practices* and *nature of science*) nor were they specific to the USA guiding documents and context (as it is the case of the *crosscutting concepts* and *core ideas*).

We started by searching for scientific studies that identified these aspects. To the authors' knowledge, only one study [65] has looked for opportunities to explore SSI in European science curricula in the early years of education, namely the Portuguese curriculum from grades 1 to 6. However, this study only examined learning goals that potentially could be used as examples of focal SSI, and not the other learning goals explored in the model for SSI teaching and learning proposed by Sadler et al. [16]. This highlights the importance of further developing this framework to include other SSI specific aspects.

The development of the subcategories in each category of analysis was conducted in two phases. A concise overview of the different phases involved in the development of the framework can be seen in Fig 1.

**Phase I: Setting and refining the subcategories within each category.** The subcategories of the category *awareness of the issue* were derived through a combined deductive-inductive approach. We adapted the framework proposed by Pessoa et al. [65] as subcategories and further extended these with subcategories that emerged from an initial analysis of the curricula. The categories *socioscientific reasoning* and *socioscientific identity* were defined based on Sadler et al. [16]. For *socioscientific reasoning*, a set of subcategories were defined based on the dimensions of this learning goal described by the same authors. We applied the initial set of categories and subcategories to a diversified sample of the curricula from England, Italy and Portugal. At this stage, we included in the analysis one grade/phase per country (ranging from grades 1 to 6). For the category *awareness of the issue* we further extended the analysis to the higher grades of school education (one grade per country, ranging from the 8th to the 11th grade) and other school subjects (Art and design, Computing, Design and technology, English, Geography, History, Mathematics, Music, Physical education, Technology), to find more specific and well-defined SSI, since in the lower grades, these are mostly mentioned in a very general way (e.g., "Know how to ask questions about local environmental problems, particularly those related to water, energy, waste, air, and soil, and how to propose solutions" in the Portuguese curriculum, grade 2). In total, seven researchers independently performed this analysis: 2 for Portuguese curriculum, 2 for English

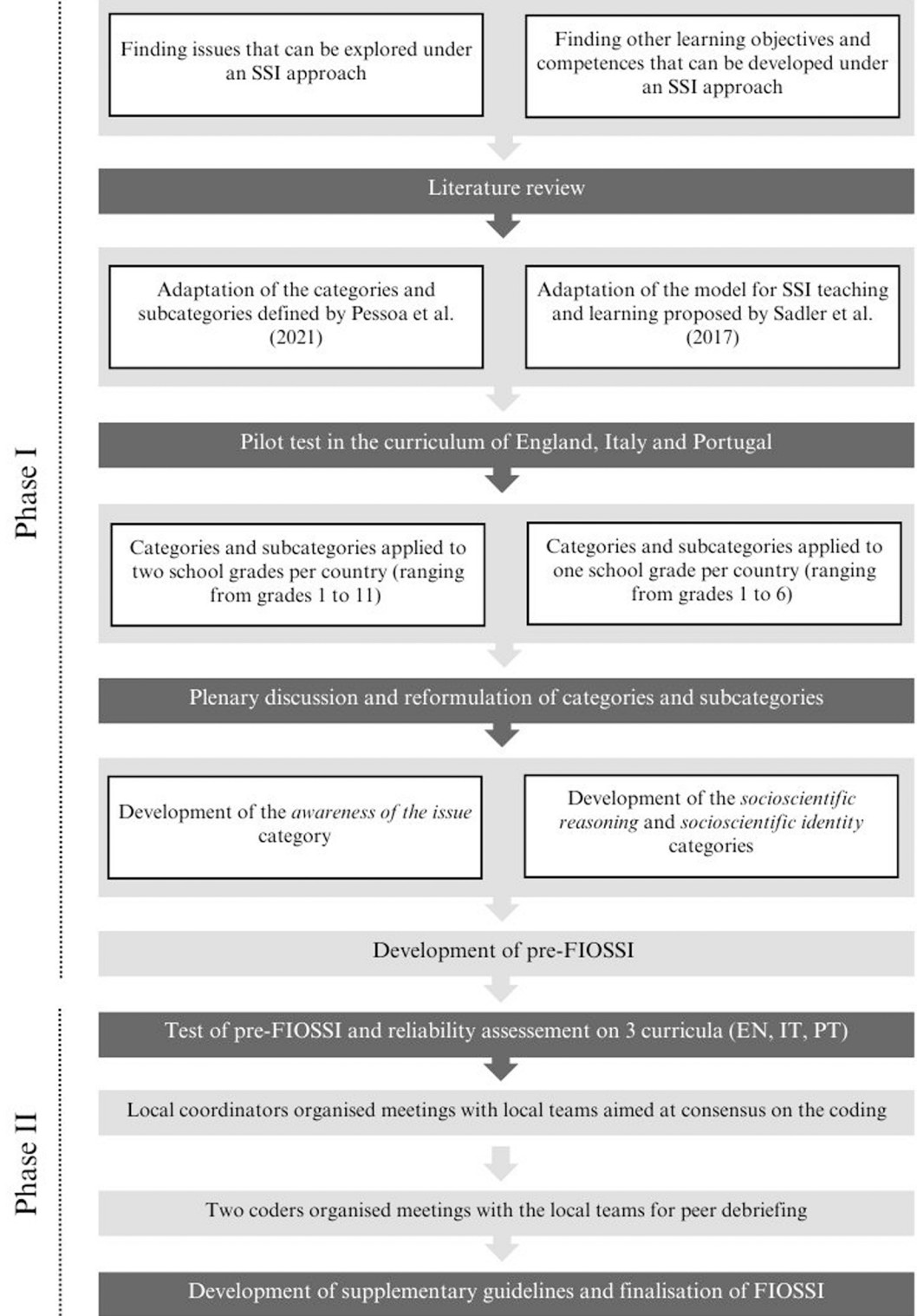

**Fig 1.  Description of the development process of FIOSSI (Framework for Identifying Opportunities to implement the SSI approach in science school curricula).**

curriculum and 3 for Italian curriculum. Researchers were asked to record each occurrence as a distinct data point. Researchers of the Italian and Portuguese curricula provided the English translation of learning goals for the international team to compare classifications and themes and ensure uniformity in interpretation [63]. The "local coordinator" in each country compiled the results of all the local coders and arranged meetings with them to review their findings and identify cases of disagreement or ambiguity. The cases found within each country's curricula were discussed and compared with those obtained by the researchers from other countries (see Fig 2 for an overview of the steps in the collaborative curriculum analysis and framework refinement process). This discussion with the entire team allowed us to identify new emergent subcategories and sub-subcategories (for *awareness of the issue*) and to revise its operational definitions (for all three categories).

**Phase II: Final development of FIOSSI and final analyses.** The framework resulting from the previous phase - pre-FIOSSI - was used to conduct the final analysis of the three science curricula. Code numbers were assigned to each subcategory, and sub-subcategory, in order to facilitate analysis. To conduct this analysis, coders independently applied FIOSSI to the school curriculum of each country. Again, coders recorded each occurrence as a distinct data point, and for the Italian and Portuguese curricula, coders provided English translations of learning goals. The percentage of agreement between the coders of each country was used as a measure of interrater reliability [66]. Two local coordinators compiled all the data and organized meetings with local teams to review their findings and identify cases of disagreement or ambiguity. These cases were discussed by the entire coding team (with the coders from the three countries; see Fig 2 for an overview of the steps in the

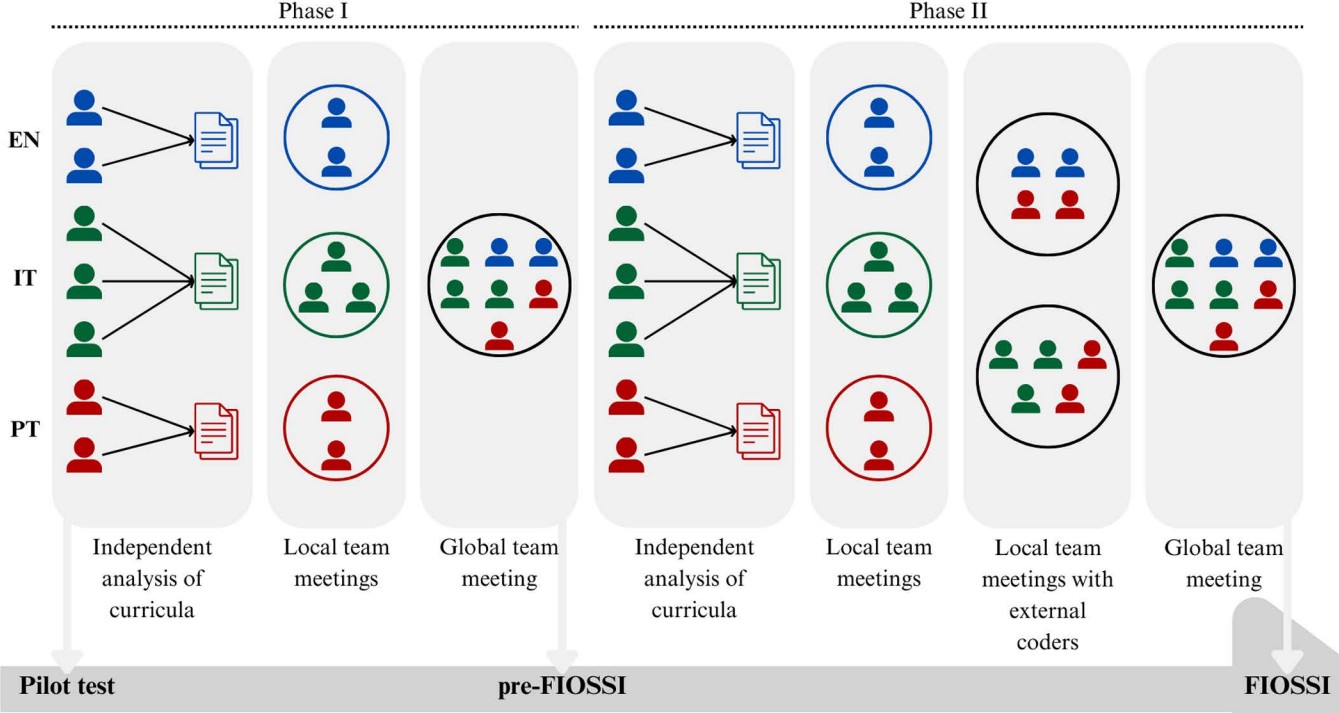

**Fig 2. Steps in collaborative curriculum analysis and framework refinement.**

collaborative curriculum analysis and framework refinement process). These meetings were used for peer debriefing to facilitate discussion of possible questions in the interpretation of learning goals, assignment of learning goals to categories and subcategories, to introduce some changes to the descriptions of categories, subcategories or sub-subcategories with lower reliability in order to resolve possible ambiguities and ensure consistency in the application of FIOSSI, and to resolve any remaining questions found by local teams to further ensure consistency of the final data across countries [63]. All coders presented their viewpoints, and the solutions incorporated everyone's perspectives. If a consensus was not reached, the learning unit was removed from the final analysis. These disagreements and discussions resulted in revisions of the analysis tool until a final version - the Framework for Identifying Opportunities to implement the SSI approach in science school curricula (FIOSSI) - was obtained (see S1 Table for the initial framework version and S2 Table for the final version).

To ensure the validity of the framework, we followed the recommendations of Cohen et al. [53]: *i)* we carried out an extensive literature review (construct validity); *ii)* we involved multiple researchers in its application and created moments of peer debriefing in the analysis of the curricula (internal validity); *iii)* we kept the subcategories and sub-subcategories not identified in the analysis of the curricula but identified in the literature review (content validity), as they may be useful in future studies.

As a result of the various phases of the analysis framework development, four general heuristics were derived regarding the application and formulation of FIOSSI:

1) general phrases, such as introductory statements or overarching goals, were incorporated into the analysis;

2) goals aimed at teacher development were excluded from the analysis;

3) where applicable, a.0 subcategory or sub-subcategory (see Fig 3) was created to enable the classification of learning objectives that are linked to the issue but which were not possible to associate with a specific subcategory or sub-subcategory;

4) specific notes for particular categories, subcategories and sub-subcategories of FIOSSI were created to clarify doubts during the coding process.

## Results

### Framework of analysis

This study aimed to develop a valid and reliable framework for teachers and researchers to identify opportunities to implement the SSI approach. The final version of the FIOSSI is depicted in Fig 3 (see S2 Table for a further presentation of the guidelines and examples for each category and subcategory). This framework comprises three categories: *awareness of the issue*, *socioscientific reasoning* and *socioscientific identity*. The first category is divided into five subcategories [1] *technology issues*, 2) *human health issues*, 3) *environmental issues*, 4) *exobiology issues*, and *social conflicts and 5) biases based on human diversity issues*) and were further divided into a total of 29 sub-subcategories. The second category is divided into six subcategories [1] *socioscientific reasoning in general*, 2) *account for the inherent complexity of SSI*, 3) *analyze issues from multiple perspectives*, 4) *identify aspects of issues that are subject to ongoing inquiry*, 5) *employ skepticism in analysis of potentially biased information*, and 6) *explore how science can contribute to the issues and understand the limitations of science in issue resolution*). The category *socioscientific identity* has no subcategories.

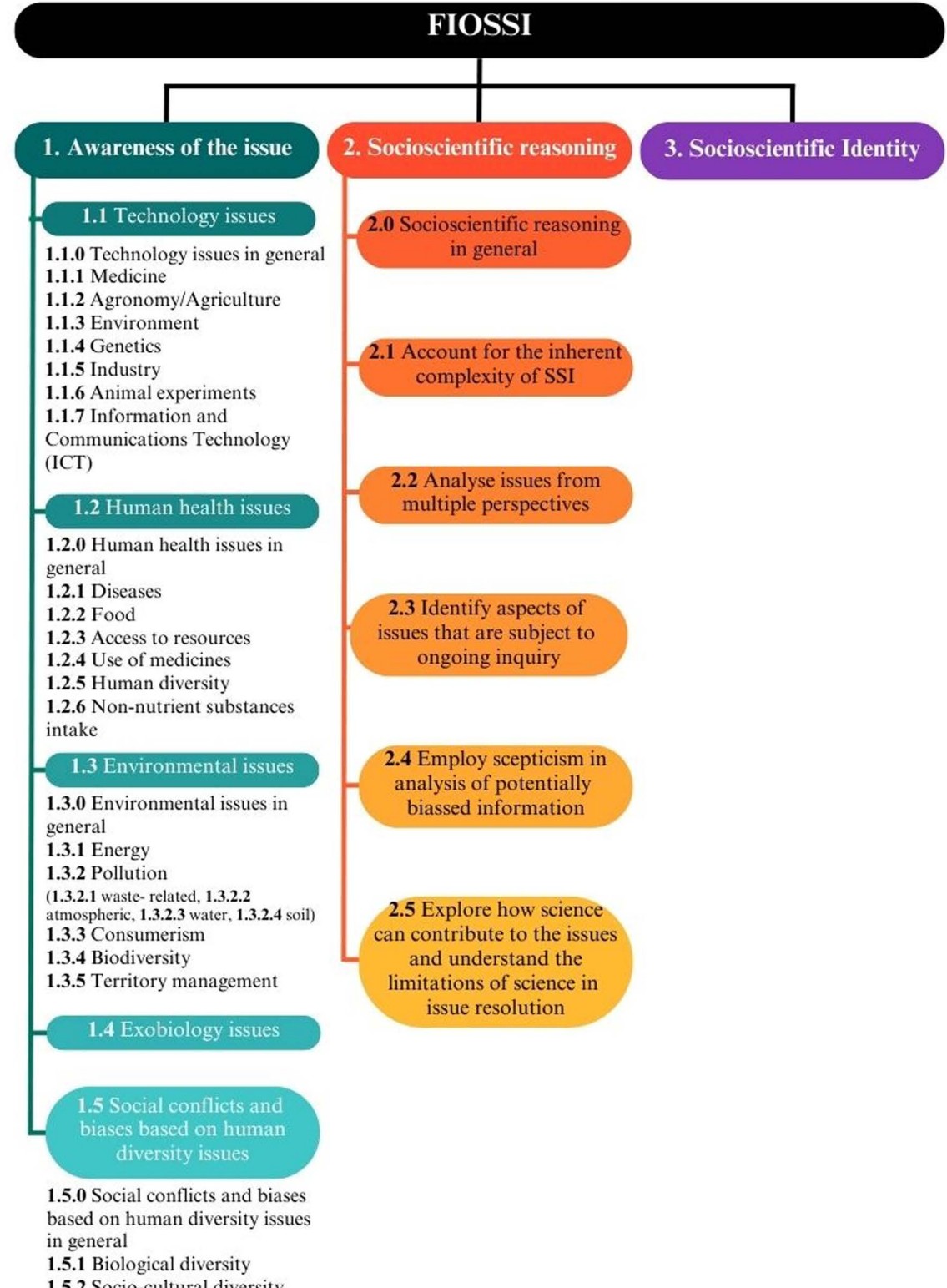

**Fig 3. Schematic representation of FIOSSI (Framework for Identifying Opportunities to implement the SSI approach in science school curricula) structure.**

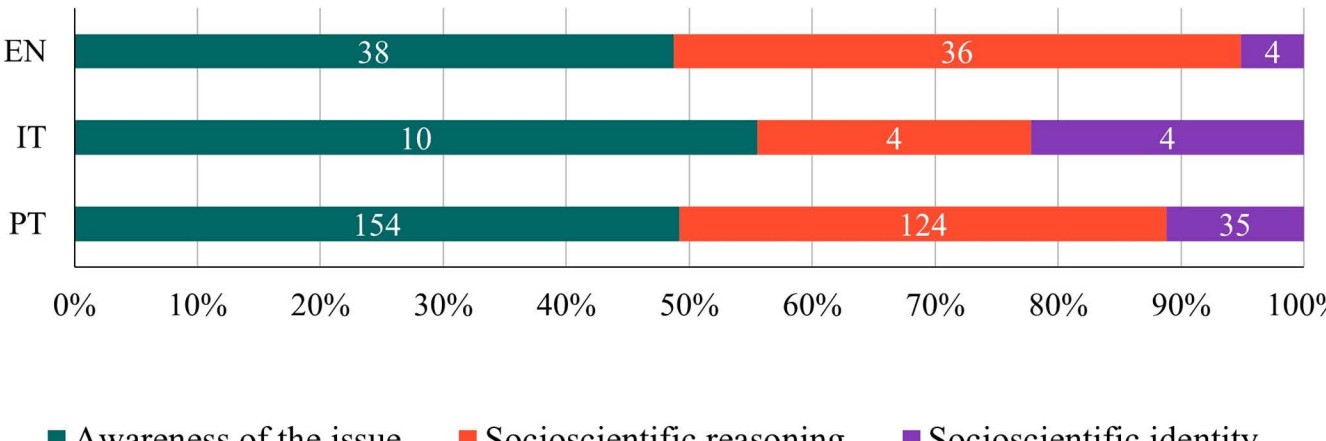

**Fig 4. Relative frequency of learning goals classified into each FIOSSI category in the English (EN), Italian (IT) and Portuguese (PT) curricula. Data labels indicate absolute frequencies.**

Modifications were also made to the initial three categories, namely as follows.

1) *Awareness of the issue*:

 a) the subcategory *biotechnology issues* retrieved from Pessoa et al. [65] was changed to become more overarching and referred to as *technology issues*; due to this change, the sub-subcategory *animal experiments* emerged; as a result of the analysis of the curricula, the sub-subcategory *information and communications technology (ICT)* was added;

 b) the subcategory *health issues* retrieved from Pessoa et al. [65] was renamed as *human health issues* to make it more accurate and, as a result of the analysis of the curricula, the sub-subcategories *human diversity* and *non-nutrient substances intake* were added;

 c) the sub-subcategory *pollution*, due to analysis of the curricula, was further divided into *waste-related*, *atmospheric*, *water* and *soil*, and the sub-subcategories *territory management* and *environmental citizenship* were added;

 d) the subcategory *social conflicts and biases based on human diversity issues* emerged from the analysis of the curricula.

2) *Socioscientific reasoning*:

 a) the previous definition of this subcategory *how science can contribute to the issues and the limitations of science* initially retrieved from Sadler et al. (16, p. 80), was changed as *explore how science can contribute to the issues and understand the limitations of science in issue resolution* to clarify its meaning and to ensure linguistic concordance with the other subcategories of the same category.

3) *Socioscientific identity*:

 a) the previous definition of the category *identity* from Sadler et al. [16] was reformulated to avoid the over-coding that occurred during the analysis in phase I; given the nature of its first guideline, '*position themselves with new competencies, interests, and ideas about*

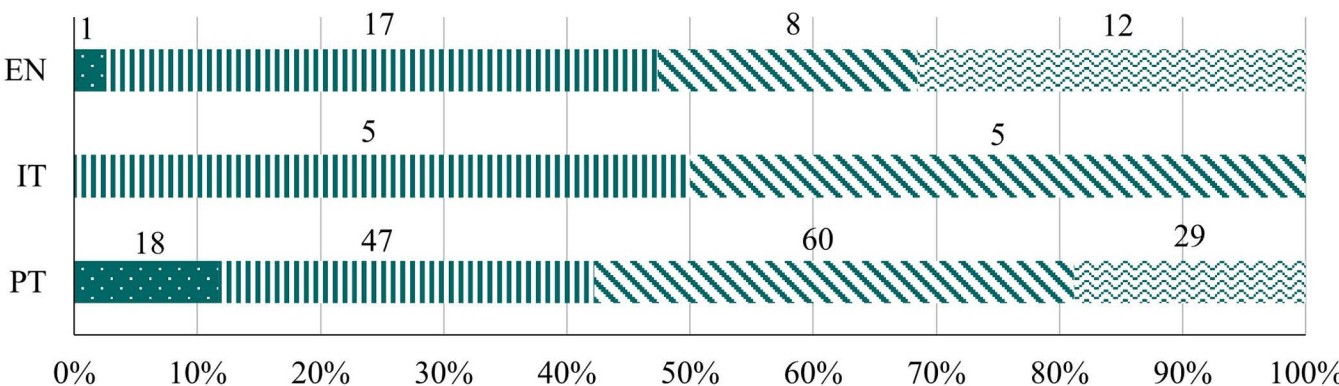

Technology issues

Human health issues

Environmental issues

Social conflicts and biases based on human diversity issues

**Fig 5. Relative frequency of learning goals classified into each *awareness of the issue* subcategory in English (EN), Italian (IT) and Portuguese (PT) curricula. Data labels indicate absolute frequencies.**

*themselves that enable new patterns of participation and discourse*' (16, p. 84), numerous learning objectives were being coded as identity since school curricula are expected to contribute to the development of new competencies in students; for example, learning goals such as EN - "The national curriculum provides pupils with an introduction to the essential knowledge that they need to be educated citizens; It introduces pupils to the best that has been thought and said"; IT - "The student is aware of the structure and development of her/his own body, in its various organs and systems, recognizes and describes their functioning, using intuitive models and takes care of her/his health"; and PT - "Acquire knowledge of oneself, developing attitudes of self-esteem and self-confidence"; since the aim of our framework is to identify opportunities to implement SSI, the name of this category was changed to *socioscientific identity*, and we reformulated its first guideline to *develop the predisposition to approach SSI using socioscientific reasoning skills*; these changes aimed to highlight that, with FIOSSI, we specifically code as *identity* development that is expected to occur during an SSI exploration.

Interrater reliability, estimated at the beginning of phase II, was higher than 70% in all countries (EN - 0.84, IT - 0.74, PT - 0.71). This represents a threshold above which the methodology is considered to retrieve reliable data [67].

## Distribution of opportunities by school years

The learning goals considered to be aligned with any of the categories of FIOSSI, together with the final consensus coding can be found in the dataset available at https://doi.org/10.48527/NN0MJC. All curricula (EN, IT, and PT) demonstrated opportunities to implement the SSI approach, with an increase in frequency as the level of education advanced (see detailed

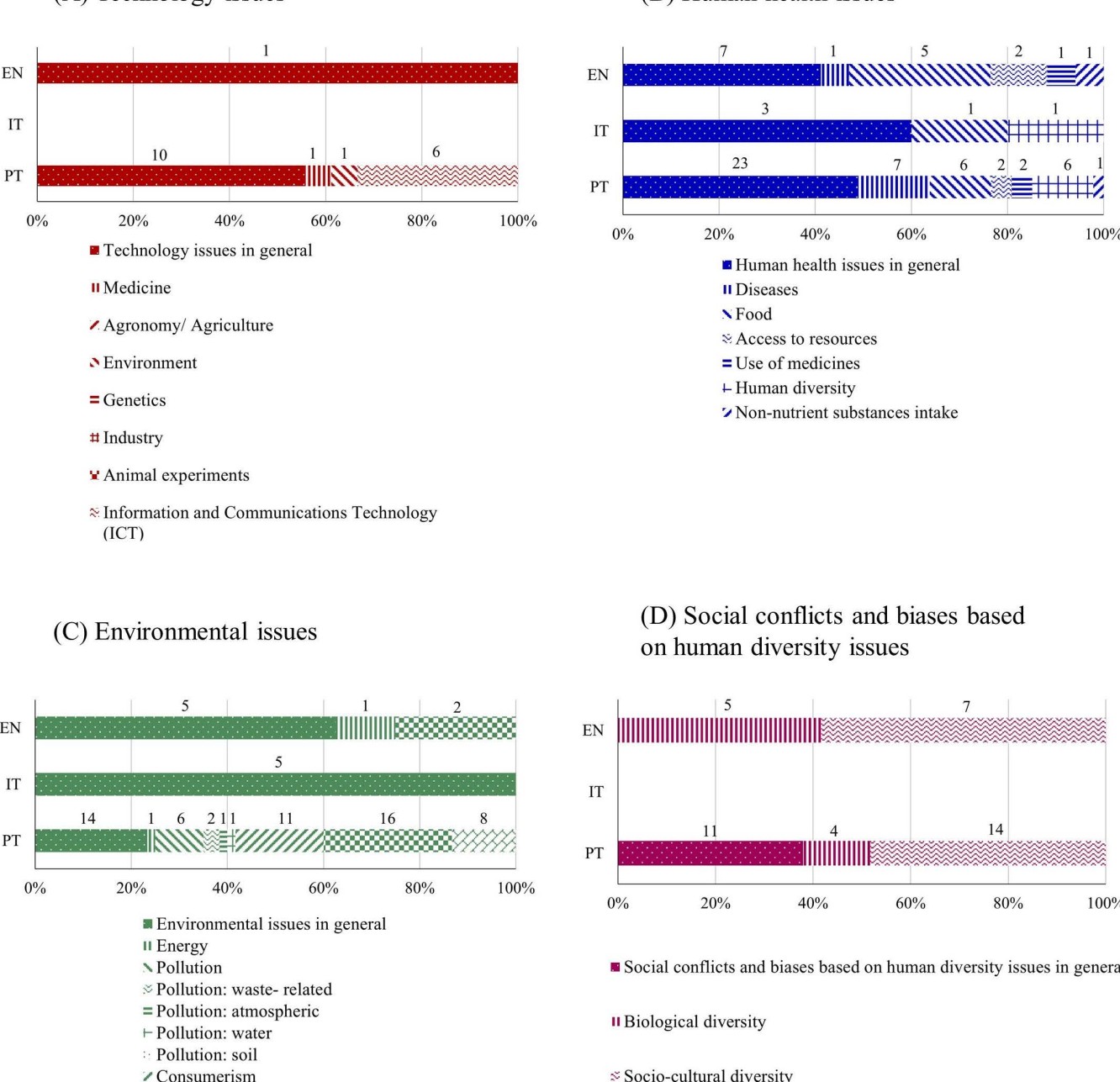

**Fig 6. Relative frequency of learning goals classified into each *awareness of the issue* sub-subcategory in English (EN), Italian (IT) and Portuguese (PT) curricula. Data labels indicate absolute frequencies.**

information in S3 Table). The IT and PT curricula showed a higher relative frequency of evidence at the higher school years analyzed (61.11% and 21.73%, respectively). The EN curriculum showed a higher relative frequency of evidence in the general aims (17.95%), however it can be observed that the highest relative frequency in specific school years was found in the upper key stage 2 (16.67%).

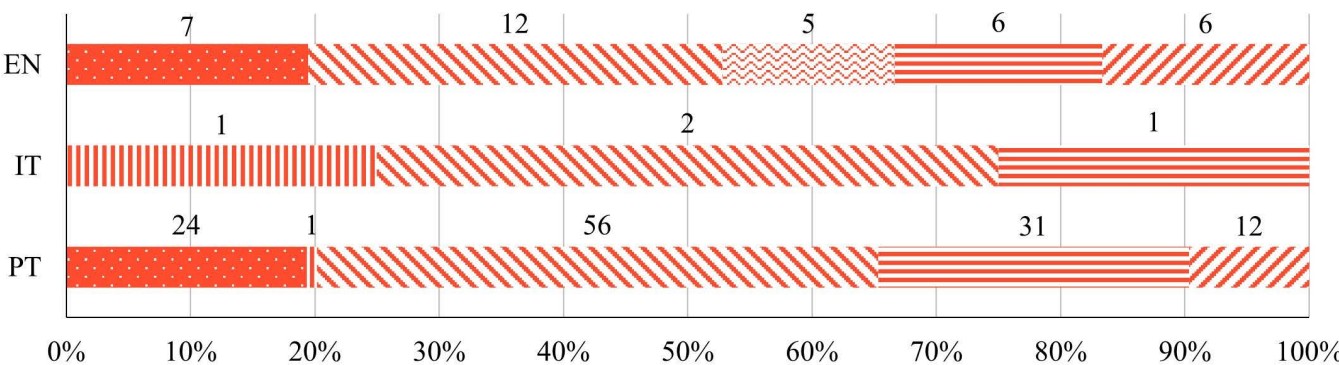

■ Socioscientific reasoning in general

‖ Account for the inherent complexity of SSI

↘ Analyze issues from multiple perspectives

≈ Identify aspects of issues that are subject to ongoing inquiry

= Employ skepticism in analysis of potentially biased information

✎ Explore how science can contribute to the issues and understand the limitations of science in issue resolution

**Fig 7. Relative frequency of learning goals classified into each *socioscientific reasoning* subcategory in English (EN), Italian (IT) and Portuguese (PT) curricula. Data labels indicate absolute frequencies.**

## Opportunities to explore the SSI approach in school science curricula

In the science curricula of the three countries included in this study, we found evidence supporting the presence of opportunities to address the three categories included in the FIOSSI. Evidence was found for the presence of all the subcategories except for *exobiology issues* (10 out of the 11 subcategories) and for 23 of the 28 sub-subcategories. No evidence was found for the sub-subcategories *agronomy/agriculture*, *genetics*, *industry*, and *animal experiments* of the subcategory *technology issues*, nor for the sub-subcategory *soil* of the subcategory *environmental issues*. Detailed results can be found in S4 Table.

In terms of the total number of evidence found, important disparities were observed among the three curricula analyzed (EN - 78, IT - 18, PT - 313). The most frequent opportunities to explore SSI in the classroom are related to the categories *awareness of the issue*, (EN - 48.72%, IT - 55.56%, PT - 49.20%, Fig 4; mostly due to issues concerning the environment and human health), and *socioscientific reasoning* (EN - 46.15%, IT - 22.22%, PT - 39,62%, Fig 4). Although relatively frequent in Italy (22.22%), the category *socioscientific identity* is not very frequent in England and Portugal (5.13% and 11.18% respectively).

**Awareness of the issue subcategories.** The subcategories *human health issues* and *environmental issues* were found in all three curricula analyzed (Fig 5). Within these subcategories, only three sub-subcategories can be found in the three curricula, namely *human health issues in general*, *food* and *environmental issues in general* (Figs 6B and 6C) and PT and EN share a higher number of sub-subcategories (namely *diseases*, *access to resources*,

*use of medicine*, *non-nutrient substances intake*, *energy* and *biodiversity*). Only the sub-subcategory *human diversity* in relation to health problems was shared by the curricula from PT and IT.

The subcategories *technology issues* and *social conflicts and biases based on human diversity issues* were only found in the EN and PT curricula (Fig 5) that share the sub-subcategories *technology issues in general*, *biological diversity* and *socio-cultural diversity* (Figs 6A and 6D). Portugal always displayed a higher number of sub-subcategories for all the subcategories related with the *awareness of the issues*, followed by England and Italy (20, 12 and 4 sub-subcategories respectively).

**Socioscientific reasoning subcategories.** The subcategories *analyze issues from multiple perspectives* and *employ skepticism in analysis of potentially biased information* were found in all three curricula analyzed (Fig 7), the former one being the most frequent (EN - 33%, IT - 50%, PT - 45% of the occurrences in this category). The subcategories *socioscientific reasoning in general* and *explore how science can contribute to the issues and understand the limitations of science in issue resolution* were only found in the EN and PT curricula. The subcategory *account for the inherent complexity of SSI* was only found in the IT and PT curricula. The subcategory *identify aspects of issues that are subject to ongoing inquiry* was only found in the EN curriculum.

From the subcategories for which evidence was found, the least frequent in the curricula were the subcategories *identify aspects of issues that are subject to ongoing inquiry* in EN (14%), *account for the inherent complexity of SSI* in PT (1%) and *account for the inherent complexity of SSI* and *employ skepticism in analysis of potentially biased information* in IT (25% each).

## Discussion

In this study, we developed a valid and reliable framework that is aligned with the literature in the field to analyze curriculum content for SSI opportunities and identified these opportunities in the science curricula of three European countries (England, Italy and Portugal). Our results showed that the three curricula analyzed have the following common characteristics:

- high frequency of opportunities to implement the SSI approach related to the *awareness of the issue*, specifically with *human health* and the *environment*;

- high frequency of opportunities to develop *socioscientific reasoning*, namely *analyzing issues from multiple perspectives* and *employing skepticism in the analysis of potentially biased information;*

- low frequency of learning goals addressing *technology issues*.

In contrast, the Italian curriculum lacks learning goals addressing *social conflicts and biases based on human diversity issues,* and shows a higher frequency of opportunities to develop *socioscientific identity* than the other two countries' curricula.

### Framework development

To the best of the authors' knowledge, this is the first attempt to create an analysis framework to identify opportunities to implement the SSI approach in the science curriculum. The application of the FIOSSI framework to three European countries' curricula with different designs and traditions showed reliable results, supporting its validity.

All the categories, and the majority of subcategories and sub-subcategories of FIOSSI were found in the analysis of the three school curricula. We did not find one subcategory and five sub-subcategories in the analysis, however decided to still keep them in the framework as they resulted from a literature review [65] and they might be found in the curricula of other

countries or in higher school grades. Garrecht et al. [68], for example, found opportunities to explore *animal experiments* in the UK and Germany curricula for secondary level. It is important to notice that the application of this framework to other countries' curricula and other school levels may reveal the need to redefine and/or include additional categories, sub-categories or sub-subcategories.

A limitation of our framework is that it does not include categories related to the development of scientific practices and the epistemology of science/nature of science, as these are not specific to the SSI approach. Learning goals addressing scientific practices and nature of science were found for these countries' curricula [69,70]. Past research, also shown how nature of science understandings can be instantiated with the SSI framework [71–76]. These studies thus suggest that in future research, the FIOSSI instrument may be extended to include categories addressing scientific practices and nature of science as these dimensions may provide additional opportunities to explore SSI.

FIOSSI is important as a tool for education research and practice and to inform education policies. Educators may use FIOSSI to identify possible topics and learning goals that can be explored through an SSI approach in their classroom. However, its usefulness for educators and effectiveness need to be tested by future studies focused on this question. When applied to multiple curricula, FIOSSI provides researchers and teachers crucial information on the learning goals that are shared across regions/countries and can be targeted for the development of educational activities and materials aligned with multiple curricula, contributing to overcome the scarcity of educational materials to address SSI [41,42]. The alignment of the activities with the curricula is expected to diminish the time-related constraints identified by teachers [37], facilitating the implementation of the SSI approach in the classrooms and contributing to foster students' understanding of the complex interplay between science, society and ethics [7] in agreement with the Visions II and III of scientific literacy [9,10,13–15].

FIOSSI can also be used by researchers to identify learning dimensions that require further study, contributing to address the current lack of information about the implementation of SSI in elementary schools and how these impact students' learning outcomes [38,39]. This information can be used to inform changes needed in curricula documents for different countries, and/or support the development of specific guidelines to incorporate SSI more effectively. In addition, the application of FIOSSI could provide further insights if extended to curricula of higher grade levels. Future research in this direction would be both valuable and impactful for broadening the framework's utility.

## Contributions to education practices and research

Within the grade ranges considered, we found that the analyzed curricula offer more opportunities to implement the SSI approach at higher school grades. This aligns with the intended purpose stated in the general objectives of all the analyzed curricula, which aim to facilitate a progressive understanding of scientific knowledge and concepts [50–52]. These results contrast with the degree of discomfort expressed by the teachers regarding the implementation of an SSI approach, which was lower in elementary school teachers [39] than in secondary school teachers [41]. In a way, our findings borders on the ironic since there is generally more openness to explore transdisciplinary topics at the elementary and middle school grade levels in comparison to high school grade levels which are often perceived to be more structured [77,78].

The relative frequency of opportunities in each category varies slightly between countries. This variation may emphasize the different approaches and priorities adopted in the educational settings of each country. However, our results also highlight opportunities to implement the SSI approach that are shared between the three countries, which are discussed below.

**Awareness of the issue.** The subcategories *environmental issues* and *human health issues* had the highest relative frequency in the three countries, suggesting the importance of teachers and researchers to focus their efforts on developing and studying activities and materials that explore such issues. Within *environmental issues*, activities focused on *biodiversity* and *energy* issues might have more potential for adaptation and dissemination across countries, as these issues are common to the English and Portuguese curricula.

We found six studies that report SSI approaches to be used in elementary education to explore *environmental issues,* exploring topics such as *energy*, *pollution* (*water pollution* and *waste-related pollution*), *consumerism*, *biodiversity* and *territory management* [38,39,79–82]. For *human health,* we found one study focusing on *diseases* [38] and one related to *food* [83]. The studies that explore *food* issues may have greater potential for adaptation since this sub-subcategory is common to the three curricula analyzed. Some educational investigations may be adapted from those available in the literature focusing STS (Science, Technology and Society) and STSE (Science, Technology, Society and Environment) approaches in science education, two movements that have been quite strong in the last decades [12]. The same applies to studies and activities focusing environmental education and health education, which a quick search on the database "Web of Science" shows to be frequently addressed in research (252 and 587 publications containing the search terms "elementary school" together with "environmental education" or "health education" respectively) but not explicitly developed under an SSI perspective (0 publications found when the term "socioscientific issues" is added to the previously reported searches). These results support the need to promote more research and the development of resources that explore *environmental* and *health issues* through an SSI approach in elementary schools.

Our results also show opportunities to explore *social conflicts and biases based on human diversity issues* in English and Portuguese curricula. However, studies exploring this topic under an SSI approach up to the 9th grade are scarce [see [84] for a notable exception], supporting the need to develop resources and research studies focused on this important issue. Higher frequencies of the sub-subcategories *issues in general* (e.g., *technology issues in general*, *human health issues in general*, *environmental issues in general*), were observed in the three analyzed curricula, suggesting that activities developed for these issues may still be useful and adaptable across countries.

**Socioscientific reasoning.** All the subcategories of *socioscientific reasoning* were found in our analysis, although none of the school curricula contained evidence for all the subcategories. However, few studies addressed or analyzed the development of *socioscientific reasoning* skills in elementary school students [85], even when we extend our search into the development of informal reasoning [defined as the scientific processes used in discussions and solutions of SSI by Sadler [86] in elementary school students [see [38,79] for notable exceptions]. This emphasizes the need for future studies in this field.

Our results reveal that, although the three school curricula provide opportunities to develop elementary school students' competencies to *analyze issues from multiple perspectives* and to *employ skepticism in analysis of potentially biased information*, very few studies were conducted in this field. This argues for the importance of developing and studying the impacts of activities that: *i)* encourage students to consider others' points of view, *ii)* evaluate counter-evidence, *iii)* analyze potential solutions from different perspectives, *iv)* recognize challenges to their own defended position, *v)* be skeptic towards potentially biased information and *vi)* develop strategies for making decisions based in credible information [87]. It is also important to develop activities focused on *socioscientific reasoning* competencies like *explore how science can contribute to the issues and understand the limitations of science in*

*issue resolution* and *account for the inherent complexity of SSI*, although these are only present in two out of the three curricula.

**Socioscientific identity.** This category was found in all the curricula analyzed, a result consistent with research suggesting that one of the key dimensions of education should be to support students as they explore and develop new identities [e.g., [88]. Identity develops over long periods of time [21]. However, it is expectable that identities evolve in response to extended efforts to engage students in meaningful negotiation of important issues [6]. Thus, although changes in students' identities are not expected after a single educational activity, they are expected to be facilitated after multiple SSI educational activities [16]. Even though a few studies have been carried out at other levels of education [e.g., [19,89], to the best of the authors' knowledge, no studies have analyzed how elementary school students develop their *socioscientific identity* through the SSI approach, or other related educational approaches that have been developed for this specific purpose.

## Implications for educational policies and curriculum developers

While our results highlight the existence of numerous opportunities to implement the SSI approach, the total number of opportunities found in the curricula of each country varies widely. These differences mirror the length of curriculum documents, the variation in the level of description of learning objectives, the number of grade levels analyzed and the organization of subjects and education cycles among the three selected countries. Thus, the comparison of such different curricular organization documents may be misleading if these differences are not carefully interpreted and considered. For example, Portugal has the most extensive curricular documents, with a very detailed level of prescription per grade, and showed a higher number of opportunities. In contrast, Italy has a less descriptive curriculum and shows a lower number of opportunities. A curriculum that presents a higher number of opportunities to explore SSI could be seen as more aligned with Visions II and III of scientific literacy. However, this deserves more careful attention and should be complemented in future studies with teachers' interviews. In fact, the length of the curriculum can be seen by teachers as imposing strong constraints, reducing teachers' willingness to engage in an SSI approach [37]. On the other hand, curricula that offer less detailed content can provide more flexibility to incorporate or deepen SSI during these specific school years, empowering schools and teachers to take responsibility and explore issues that are locally more relevant and engaging for the educational community [45,90]. However, it is essential to consider the distinction made by curriculum theorists between the formal curriculum (prescribed), the hidden curriculum (implicit), and the real curriculum (in action) [91]. Therefore, an analysis of the formal curriculum, such as the one we have conducted, is not sufficient to fully assess the alignment of school curricula with visions of scientific literacy. Moreover, in contrast with the curricula from EN and IT, the PT curriculum merges topics of Geography, History and Technology with natural sciences (Biology, Physics, Geology and Chemistry) from grades 1 to 4. This may explain the higher number and diversity of opportunities identified in Portuguese curriculum. Exploring SSI in disciplines other than the science-related discipline could affect how these issues are addressed [92]. In schools employing student-centered integrated curriculum programs, organized around significant problems and issues without regard to subject area lines, students were shown to have better academic outcomes than students in schools where no integrated curriculum approach was used [reviewed in [93,94]. This suggests that exploring SSI in non-science disciplines can affect how science is used to reason about these issues, but exploring SSI only in science disciplines can also limit other perspectives besides the scientific one. Future studies on the impacts of an SSI pedagogical approach in interdisciplinary or transdisciplinary environments compared to single-disciplinary environments will be worthwhile. We should

highlight that the inclusion of other disciplines in the analysis, such as Technology, History, Geography and Citizenship, could contribute to a broader vision and a more comprehensive understanding of the range of opportunities to address SSI.

Despite the considerations above, our results highlight features in the elementary school science curriculum of each country that deserve further attention and which are discussed below.

**Awareness of the issue.** In this category, the subcategory *technology issues* was one of the least covered by the English and Portuguese curricula and was not identified in the Italian curriculum (although we should note that, in the three analyzed countries, Technology is a separate subject). With technology playing such a significant role in our daily lives today, not addressing technology-related issues within science education may limit students' preparation to effectively navigate the ethical, social, and environmental implications and challenges associated with technology and the relationship between science and technology [95,96]. The limited emphasis on technology at lower levels of education can impact the development of the skills and knowledge that students need to critically engage with SSI, and teachers' competence and confidence to integrate technology into their science teaching practices [96].

The IT curriculum does not include learning goals related with *social conflicts and biases based on human diversity issues.* However, introducing social conflicts and biases associated with both biological and socio-cultural diversity in science teaching may provide opportunities to promote awareness about inequalities prevalent in society, to develop a deeper understanding of the complexities surrounding diversity and to actively contribute to decreasing inequality and discrimination in line with the goals for the 2030 Agenda of the United Nations [1,97,98]. More specifically, exploring these issues may contribute to education for sustainability [4,5] by addressing gender equality and empowerment of all women and girls (SDG5), strategies to reduce inequalities (SDG10), and to promote peaceful and inclusive societies for sustainable development, providing access to justice for all and building effective, accountable and inclusive institutions at all levels (SDG16).

**Socioscientific reasoning.** None of the curricula analyzed presented opportunities to develop all subcategories of the socioscientific reasoning, thus compromising students' opportunities to develop important competencies to fully understand and address SSI. More specifically, the English curriculum does not explicitly refer to opportunities for students to *account for the inherent complexity of SSI*, thus increasing the probability of failing to recognize multiple and dynamic interactions within the issue. Additionally, the Italian and Portuguese curricula do not explicitly refer opportunities for students to *identify aspects of issues that are subject to ongoing inquiry* what may compromise students' ability to recognize that SSI are real problems, characterized by a degree of uncertainty, and ill-structured with underlying assumptions, conditions and potentially significant information not always determined or known. Furthermore, the Italian curriculum does not explicitly mention opportunities for students to *explore how science can contribute to the issues and understand the limitations of science in issue resolution*, which may compromise the development of students' scientific literacy. In fact, independently of the issue addressed, all these competencies play an essential role in the development of scientific literacy Vision III [8], as all of them emphasize the integration of considerations beyond scientific knowledge, such as social and ethical ones, and encourage the critical analysis of real problems. Since SSI are often related to sustainable development [99,100], the implementation of the SSI approach also provides opportunities to address many of the issues outlined in the SDGs and emphasizes the integrated and indivisible nature of the world's challenges. Moreover, the key competencies in sustainability [4], defined as essential for individuals to transform their lifestyles and to contribute to societal transformation towards sustainability, share

common elements with socioscientific reasoning competencies. These common elements include: 1) a focus on preparing students to understand, engage with, and contribute to a more sustainable and equitable world; 2) an emphasis on critical thinking (critical thinking competency/ *employ skepticism in analysis of potentially biased information*); inclusion of interdisciplinary perspectives (systems thinking competency/ *account for the inherent complexity of SSI* and *analyze issues from multiple perspectives*); 3) ethical considerations (normative competency/ *analyze issues from multiple perspectives* and *explore how science can contribute to the issues and understand the limitations of science in issue resolution*); and, 4) attention on addressing complex-real world issues [4,16]. This further supports the importance of ensuring that all the dimensions of socioscientific reasoning are explicitly mentioned in the curricula for these to actually contribute to the promotion of education for sustainability.

**Socioscientific identity.** In the English and Portuguese curricula, *socioscientific identity* was the least frequently addressed category. These results may be related to the underlying nature of this category. Since the development of *socioscientific identity* entails the development of students' ability to recognize themselves as active participants in the resolution of SSI, empowering them to engage in discussion, decision-making, and problem-solving [16], these learning objectives may be also present in subjects other than science. For example, all three countries have specific subjects related to citizenship education. In England, the citizenship subject only starts from grade 7 (11 years old) and its purpose is to: "equip pupils with the skills and knowledge needed to critically explore political and social issues, to weigh evidence, debate and make reasoned arguments... It should also prepare pupils to take their place in society as responsible citizens…" (51, p.227). In Italy, the subject of "Citizenship and Constitution" is transversal and compulsory in all grades, and aims to build "a sense of legality and the development of an ethic of responsibility, which materialize in the duty to choose and act consciously, and which implies the commitment to develop ideas and promote actions aimed at the continuous improvement of the environment in which one lives, starting with daily life at school and personal involvement in usual routines…" (52, p. 27). In addition, since 2020, the subject "Civic education" has been added to the curriculum, covering sustainable development, environmental education and digital citizenship [101]. In Portugal, "citizenship and development" is also a transversal and compulsory subject in all grades and has the purpose to "contribute to students' fulfillment through the full development of their personality, attitudes and sense of citizenship. In this way, students are prepared to consciously reflect on spiritual, aesthetic, moral and civic values in order to ensure their balanced civic development." (53, p. 2). Given the purposes of citizenship education, analyzing this subject may highlight additional opportunities to develop students' *socioscientific identity* and for implementing the SSI approach in these countries' curricula.

Not acknowledging opportunities to develop *socioscientific identity* in science curricula can compromise the full development of scientific literacy Vision III, since students might not have other opportunities to recognize their active role as citizens in science-related issues, which is an aspect mentioned in the definition of scientific literacy by various professional organizations [9,10]. Moreover, a higher number of opportunities to develop *socioscientific identity* could foster the development of key competencies in sustainability such as strategic and self-awareness competencies, defined as "the ability to collectively develop and implement innovative actions that further sustainability at the local level and further afield" (4, p. [102]), and "the ability to reflect on one's own role in the local community and (global) society, continually evaluate and further motivate one's actions, and deal with one's feelings and desires" (4, p. [102]), respectively. It is, therefore, important that socioscientific identity is clearly and often addressed in the science curricula and explored in the classrooms.

### Implications for elementary teacher training and further research

The successful implementation of the SSI approach in the classroom relies on the teachers. In this sense, curriculum analyses represent a simplistic view of what actually takes place in classrooms. Although curricula mandates the legal obligations of schools and teachers in terms of what they must teach [45], many other factors influence teachers' practices, including their pedagogical beliefs [8]. However, few studies analyzed the implementation of SSI by elementary school teachers [37,39]. Therefore, further studies on elementary teachers' practices (if, how and when they implement the SSI approach) and on the educational resources they use, are needed to more accurately identify opportunities to explore SSI in the classroom and develop effective educational resources.

The identification of opportunities to implement the SSI approach can serve as a valuable guide for directing efforts for both in-service and pre-service teacher training to overcome the lack of confidence in SSI implementation reported by pre-service elementary teachers [37]. Given the limited number of studies, further research is needed to identify effective methodologies to empower teachers to develop, adapt and implement resources and pedagogical strategies to facilitate the SSI implementation.

## Summary

Our study provides the first framework for identifying opportunities to implement the SSI approach in countries' curricula. Our application of FIOSSI to the science curricula of England, Italy, and Portugal has yielded results that demonstrate that the three countries share common learning goals in their elementary school curricula. Specifically, these goals encompass addressing topics related to *environmental issues* and *human health issues*, to develop skills such as *analyzing issues from multiple perspectives* and *employing skepticism in the analysis of potentially biased information*, and to develop *socioscientific identity*. This suggests that activities developed to address these learning goals through an SSI approach may be used and adapted in the three countries. We also show that there are important issues in the three curricula that deserve further attention and may inform policy changes. These issues include the low frequency of learning goals addressing *technology issues* in the science curricula, and the absence of learning goals addressing *social conflicts and biases based on human diversity issues* in the Italian curriculum. We also argue for the importance of including learning goals covering all subcategories of *socioscientific reasoning* and increase the frequency of those addressing the development of *socioscientific identity* in the curricular documents of the three countries. Our results represent an important contribution to develop solutions to improve students' scientific literacy and sustainability key competencies and may serve as a catalyst for educational reform and the advancement of science education towards a more holistic and socially relevant approach, ultimately, contributing to the SDGs.

## Supporting information

**S1 Table. Initial framework of analysis**
(DOCX)

**S2 Table. Framework for Identifying Opportunities to implement the SSI approach in science school curricula (FIOSSI). \* The examples provided are learning objectives from the analyzed curricula.** EN - English school curriculum, IT - Italian school curriculum, PT - Portuguese school curriculum. IT and PT learning objectives were translated by the authors. [i] Subcategories and sub-subcategories that emerged from the inductive analysis.
(DOCX)

**S3 Table. Distribution of opportunities for SSI found in the three curricula analyzed according to the structure of each curriculum** fr (%) - relative frequency within each country (English - N = 78, Italian - N = 18, Portuguese - N = 313, with N meaning the total number of evidence found in each country)
(DOCX)

**S4 Table Absolute frequencies of the learning goals attributed to a FIOSSI category, subcategory and sub-subcategory per school curriculum (see the definition of FIOSSI categories, subcategories and sub-subcategories in S2 Table)**
(DOCX)

## Author contributions

**Conceptualization:** Patrícia Pessoa, Joelyn de Lima, J. Bernardino Lopes, Alexandre Pinto, Xana Sá-Pinto.

**Data curation:** Patrícia Pessoa, Joelyn de Lima, Valentina Piacentini, Giulia Realdon, Alex Jeffries, Lino Ometto, Xana Sá-Pinto.

**Formal analysis:** Patrícia Pessoa.

**Funding acquisition:** Patrícia Pessoa, Xana Sá-Pinto.

**Investigation:** Patrícia Pessoa, Joelyn de Lima, Valentina Piacentini, Giulia Realdon, Alex Jeffries, Lino Ometto, Xana Sá-Pinto.

**Methodology:** Patrícia Pessoa, Joelyn de Lima, Valentina Piacentini, Xana Sá-Pinto.

**Project administration:** Patrícia Pessoa, Xana Sá-Pinto.

**Resources:** Patrícia Pessoa, Joelyn de Lima, Valentina Piacentini, Giulia Realdon, Alex Jeffries, Lino Ometto.

**Supervision:** J. Bernardino Lopes, Dana L. Zeidler, Alexandre Pinto, Xana Sá-Pinto.

**Validation:** Patrícia Pessoa, Xana Sá-Pinto.

**Visualization:** Patrícia Pessoa.

**Writing – original draft:** Patrícia Pessoa.

**Writing – review & editing:** Patrícia Pessoa, Joelyn de Lima, Valentina Piacentini, Giulia Realdon, Alex Jeffries, Lino Ometto, J. Bernardino Lopes, Dana L. Zeidler, Maria João Fonseca, Bruno Sousa, Alexandre Pinto, Xana Sá-Pinto.

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
