## [Decision Letter · Decision Letter 0]

27 Sep 2024

PONE-D-24-32159

A Framework to Identify Opportunities to Address Socioscientific Issues in the Elementary School Curricula: A Case Study from England, Italy, and Portugal

PLOS ONE

Dear Dr. Pessoa,

Thank you for submitting your manuscript to PLOS ONE. After careful consideration, we feel that it has merit but does not fully meet PLOS ONE’s publication criteria as it currently stands. Therefore, we invite you to submit a revised version of the manuscript that addresses the points raised during the review process.

We look forward to receiving your revised manuscript.

Kind regards,

Tahir Turk, PhD

Academic Editor

PLOS ONE

Journal Requirements:

3. Please note that your Data Availability Statement is currently missing the repository name and/or the DOI/accession number of each dataset OR a direct link to access each database. If your manuscript is accepted for publication, you will be asked to provide these details on a very short timeline. We therefore suggest that you provide this information now, though we will not hold up the peer review process if you are unable.

Additional Editor Comments:

Specifically, please address the issues as follow (see reviewer feedback below_:

Issues on consistency of sampling and comparative benchmarks of documents from the different countries. This includes more detail on the principles, protocols, and considerations for subsequent application of this analytical framework.

Reviewers' comments:

Reviewer's Responses to Questions

**Comments to the Author**

1. Is the manuscript technically sound, and do the data support the conclusions?

Reviewer #1: Yes

2. Has the statistical analysis been performed appropriately and rigorously? 

Reviewer #1: Yes

3. Have the authors made all data underlying the findings in their manuscript fully available?

Reviewer #1: Yes

4. Is the manuscript presented in an intelligible fashion and written in standard English?

Reviewer #1: Yes

5. Review Comments to the Author

Reviewer #1: Developing the Framework for Identifying Opportunities to implement the SSI approach in school science curricula is an importand and contributing researh topic that can be applied to the comparative analysis of more cross-national curriculum documents in the future. In view of the potential of this framework, the authors can provide more introduction or discussion on the positioning of curriculum documents in various countries (such as guidelines for textbook compilation and practical teaching). From the reports in this study, we have noticed that the number of pages in curriculum documents in the three countries varies greatly, which may also reflect differences in the focus and details of the documents. Furthermore, for the category awareness of the issue, the analysis materials must be extended to grades 8-11 and even other school subjects. I would like to know how the sampling and comparative benchmarks of the documents from different countries can be consistent. Therefore, it is recommended that the authors provide relevant information on the principles, protocols, and considerations for subsequent application of this analytical framework.

6. PLOS authors have the option to publish the peer review history of their article (what does this mean? ). If published, this will include your full peer review and any attached files.

**Do you want your identity to be public for this peer review?** For information about this choice, including consent withdrawal, please see our Privacy Policy .

Reviewer #1: No

---

## [Author Response · Author response to Decision Letter 1]

11 Nov 2024

Response to Reviewers

We truly appreciate all the comments and suggestions from the editor and reviewer. We believe that all of them have greatly contributed to improve our work. We present below all the changes made to address the concerns and requests received. We hope we have successfully incorporated all recommendations and that our paper can now be considered for publication.

On behalf of the authors’

Patrícia Pessoa

Reviewer Comments:

Reviewer #1: Developing the Framework for Identifying Opportunities to implement the SSI approach in school science curricula is an important and contributing research topic that can be applied to the comparative analysis of more cross-national curriculum documents in the future.

Thank you for your kind words. We hope that the changes made in the paper address the reviewer’s main concerns.

In view of the potential of this framework, the authors can provide more introduction or discussion on the positioning of curriculum documents in various countries (such as guidelines for textbook compilation and practical teaching).

Thank you for this valuable suggestion. To address your suggestion, we have expanded the text to provide further context on the positioning of curriculum documents in relation to practical teaching. Specifically, we added a sentence in our sample section highlighting the commonality across the three curricula while also noting the differences between them:

“We focused on the analysis of elementary school science curricula, which share common features, such as the inclusion of learning objectives and suggestions for teaching strategies, yet show wide variation in terms of organization (see Table 1) and flexibility provided to teachers and disciplines. Besides the learning goals that the students are expected to achieve, these documents also provide guidelines for educational approaches to be implemented by the teachers.”

Furthermore, we clarified the requirements for textbook alignment with curriculum standards in each country:

“According to the national regulations in Italy and Portugal, textbooks need to comply with the objectives, contents, and guidelines in the official curriculum [Italy: (55); Portugal: (56,57)]. In England, there is no specific legislation requiring textbooks to meet curriculum standards; however, publishers typically ensure alignment to support schools in meeting these requirements.”

Note: Due to these changes, we have updated our reference list to include these citations.

From the reports in this study, we have noticed that the number of pages in curriculum documents in the three countries varies greatly, which may also reflect differences in the focus and details of the documents.

Thank you for your insightful observation. We agree that the differences in page count across curriculum documents reflect indeed variations in focus and in the level of detail within each curriculum. As we recognise already in the discussion section:

“While our results highlight the existence of numerous opportunities to implement the SSI approach, the total number of opportunities found in the curricula of each country varies widely. These differences mirror the length of curriculum documents, the variation in the level of description of learning objectives, the number of grade levels analyzed and the organization of subjects and education cycles among the three selected countries. Thus, the comparison of such different curricular organization documents may be misleading if these differences are not carefully interpreted and considered.”

But, to make this clear sooner, besides acknowledging the differences between the curricula in our sample section, we now changed a sentence to highlight that:

“(...) it is not the aim of this study to compare the curricula of the three countries, but rather to highlight the opportunities for addressing socioscientific issues within each curriculum individually(...)”

Furthermore, for the category awareness of the issue, the analysis materials must be extended to grades 8-11 and even other school subjects.

We sincerely appreciate your insightful comment and the suggestion to expand the analysis of the "awareness of the issue" category to include higher grades and additional subjects. We acknowledge the value of this approach and the importance of addressing socioscientific issues across educational levels. However, we would like to clarify that this study specifically focuses on analyzing opportunities to address socioscientific issues within elementary school curricula. This choice was made to address a gap in the literature, as stated in the introduction:

“While there are some promising studies that have shown SSI-based education to be particularly relevant for elementary school students connecting science knowledge with social issues, thereby providing a foundation for more complex reasoning and understanding of scientific concepts (36–39), most of the research and resources in the extant literature tend to be focused on higher education, not on elementary school levels (36–39).”, and “To the best of the authors' knowledge, to date, no study has investigated opportunities to implement SSI at the elementary school level in European science school curricula.”

That said, we recognize the relevance of your suggestion, and we have now included this idea in our discussion as a potential direction for future research:

“In addition, the application of FIOSSI could provide further insights if extended to curricula of higher grade levels. Future research in this direction would be both valuable and impactful for broadening the framework's utility.”

Regarding the inclusion of other subjects, we agree that the analyisis of SSI-related learning objectives in fields beyond science would be very interesting. However, as mentioned in our sample section, this study focused solely on science subjects:

“While there may be relevant SSI-related learning goals in subjects like History or Technology, our analysis focused solely on goals taught within subjects in the field of natural sciences.”

In our discussion section, we also propose this as a suggestion for future work:

“We should highlight that the inclusion of other disciplines in the analysis, such as Technology, History, Geography, and Citizenship, could contribute to a broader vision and a more comprehensive understanding of the range of opportunities to address SSI.”

I would like to know how the sampling and comparative benchmarks of the documents from different countries can be consistent.

We appreciate the reviewer’s interest in our approach to document selection and analysis across countries. We would like to clarify that our objective is not to conduct direct comparisons between countries, as the specific subjects and curricula differ in scope and structure. Instead, our study aims to identify and highlight the opportunities for addressing socioscientific issues within each curriculum individually. However, we were attentive in our selection of grade levels and subjects. Regarding the subjects, as previously mentioned, we focus on those in which science is explicitly included or integrated into the curriculum. In terms of grades, we focused on those taught by primary teachers in each country. That said, we did seek a degree of consistency by selecting similar/equivalent national documents for analysis across the three countries, ensuring that each reflects educational stages and primary-level science content. Given the reviewer's comment, we felt this might not be clear enough, and so we introduced the following change to the sample section:

“Although it is not the aim of this study to compare the curricula of the three countries, but rather to highlight the opportunities for addressing socioscientific issues within each curriculum individually, we focused both on comparable grade ranges in terms of years of schooling and students’ age, and teacher training”.

Therefore, it is recommended that the authors provide relevant information on the principles, protocols, and considerations for subsequent application of this analytical framework.

Thank you for your valuable suggestion. To address this, we have incorporated Fig 2 to clarify the principles, protocols, and iterative steps that guided the application of our framework. This figure provides an overview of our collaborative process, from individual analysis and local team meetings to global team discussions and refinement stages, trying to make our methodology more transparent for potential subsequent applications.

Fig 2. Steps in collaborative curriculum analysis and framework refinement

---

## [Decision Letter · Decision Letter 1]

10 Jan 2025

A Framework to Identify Opportunities to Address Socioscientific Issues in the Elementary School Curricula: A Case Study from England, Italy, and Portugal

PONE-D-24-32159R1

Dear Dr. Pessoa,

We’re pleased to inform you that your manuscript has been judged scientifically suitable for publication and will be formally accepted for publication once it meets all outstanding technical requirements.

Kind regards,

Tahir Turk, PhD

Academic Editor

PLOS ONE

Additional Editor Comments (optional):

Reviewers' comments:

Reviewer's Responses to Questions

**Comments to the Author**

1. If the authors have adequately addressed your comments raised in a previous round of review and you feel that this manuscript is now acceptable for publication, you may indicate that here to bypass the “Comments to the Author” section, enter your conflict of interest statement in the “Confidential to Editor” section, and submit your "Accept" recommendation.

Reviewer #1: All comments have been addressed

2. Is the manuscript technically sound, and do the data support the conclusions?

Reviewer #1: Yes

3. Has the statistical analysis been performed appropriately and rigorously? 

Reviewer #1: Yes

4. Have the authors made all data underlying the findings in their manuscript fully available?

Reviewer #1: Yes

5. Is the manuscript presented in an intelligible fashion and written in standard English?

Reviewer #1: Yes

6. Review Comments to the Author

Reviewer #1: The authors have revised this manuscript in response to my previous comments. I have no further comment and look forward to seeing it published.

7. PLOS authors have the option to publish the peer review history of their article (what does this mean? ). If published, this will include your full peer review and any attached files.

**Do you want your identity to be public for this peer review?** For information about this choice, including consent withdrawal, please see our Privacy Policy .

Reviewer #1: **Yes: ** Shiang-Yao Liu

---

## [Editor Report · Acceptance letter]

PONE-D-24-32159R1

PLOS ONE

Dear Dr. Pessoa,

I'm pleased to inform you that your manuscript has been deemed suitable for publication in PLOS ONE. Congratulations! Your manuscript is now being handed over to our production team.

Kind regards,

on behalf of

Dr. Tahir Turk

Academic Editor

PLOS ONE
